# High Gain Improved Planar Yagi Uda Antenna for 2.4 GHz Applications and Its Influence on Human Tissues

**Claudia Constantinescu \***, **Claudia Pacurar**, **Adina Giurgiuman, Calin Munteanu, Sergiu Andreica** and **Marian Gliga**

Department of Electrotechnics and Measurements, Faculty of Electrical Engineering, Technical University of Cluj-Napoca, 400114 Cluj-Napoca, Romania; claudia.pacurar@ethm.utcluj.ro (C.P.); adina.giurgiuman@ethm.utcluj.ro (A.G.); calin.munteanu@ethm.utcluj.ro (C.M.); sergiu.andreica@ethm.utcluj.ro (S.A.); marian.gliga@ethm.utcluj.ro (M.G.)
* Correspondence: claudia.constantinescu@ethm.utcluj.ro

**Abstract:** Considering the technological enhancements nowadays, antennas tend to be smaller in order to be easily integrated in devices. The most used antennas today in small high-tech devices close to the human body are planar antennas. In this paper, a Yagi Uda planar antenna operating at a frequency of 2.4 GHz is HF analyzed and optimized by increasing its bandwidth and gain while maintaining its initial dimensions. The methods used to optimize the antenna's operation are the use of different dielectrics, different numbers of directors, and different dimensions for directors, placing new conductor elements, all while keeping the same dimensions for its implementation on the planar device. The optimized structure of the planar Yagi Uda antenna has a 10% increase in bandwidth and a 30% increase in gain, reaching a peak value of 4.84 dBi. In our daily activities, we use devices with such antennas very often, so an analysis of the antenna's influence on the human body is performed: the SAR, electric and magnetic field and radiation power density are determined, represented and reported to the standards in force. For the frequency considered, the SAR should be below 4 W/kg for the head/torso when the exposure is more than six minutes, which is a value exceeded by the antenna in its near vicinity. The calculated maximum electric field limit is 0.349 V/m and the maximum magnetic field value is 28.441 V/m for an exposure between 6 and 30 min values, which is also exceeded in the immediate vicinity of the antenna. The results allow us to suggest that such an antenna should be placed further from the human body, or some protection should be placed between the body and the antenna. From the radiation power density point of view for the modeled antenna, it can be said that a distance from the antenna greater than 0.5 m is considered to be safe.

**Keywords:** dielectric; geometrical dimensions; S parameters; gain; SAR; electric field; magnetic field

## 1. Introduction

At present, antennas are everywhere, from TV broadcasting and mobile services to biomedical applications and sensors. Due to a lack of space and the continuous decrease in equipment dimensions, the antennas considered to be integrated must be also small and easy to place in tight spaces. That is why planar antennas are highly used in nowadays, and they are shaped to reach the goals of a higher gain and bandwidth while making the dimensions as small as possible.

There are a lot of domains in which planar antennas are used, such as biomedical applications [1], wireless local area network (WLAN) applications [2,3], monitoring changes of the dielectric constant of the atmosphere caused by the presence of pollution [4], microwave sensors to evaluate the dielectric characteristics of different types of materials [5], thermoelectric sensors [6], mapping of terrain, and many others.

Almost all the classical antennas have a planar correspondent, and in this case, this paper is considering the analysis of a planar Yagi Uda Antenna [7,8]. The first Yagi Uda

planar antenna was studied in 1998, but its configuration is improving still in the present studies. These antennas are used in different bandwidths depending on their application, from L-band [9,10], MICS (Medical Implant Communication System) and ISM (Industrial, Scientific, Medical) 2.4 GHz bands [3,9,11–13], to 5G applications [13,14] and even terahertz applications. The antenna constructed in this study is operating at a frequency of 2.4 GHz; thus, a short comparison to the other antennas created to be used in the same frequency range is necessary. Table 1 consists of the main characteristics of such antennas from recent studies. The purpose of the present research is an increased gain value and a larger bandwidth for a Yagi Uda planar antenna while maintaining the same small dimensions for the initial structure.

There are a lot of ways to increase the gain of an antenna, and some of them like using more layers of substrate, using a certain kind of shape for the patch, using an air gap in the structure or using meta lenses, as mentioned in [15,16]. In this paper, the purpose of increasing the gain was reached by modifying the geometry of the patch and ground, always evaluating the bandwidth also.

**Table 1.** Similar structures from recent studies with their characteristics.

| Reference | Antenna | Dimensions (mm × mm × mm) | Bandwidth (%) | Peak Gain (dBi) |
|---|---|---|---|---|
| [3] |  | 55 × 48 × 1.6 | 8% | 4.34 |
| [8] |  | 76 × 86 | 53.5% | 4.65 |
| [9] |  | 100 × 98 × 0.6 | 7.4% | 7.09 |
| [13] |  | 68 × 68 × 0.8 | 15% | 3.05 |
| [17] |  | 83.63 × 184 × 1.5 | 28.6% | 5.7 |
| [18] |  | 42 × 20 × 0.25 | 13% | 4.1 |

With the increasing number of networks, telecommunication manufacturers and operators need to be more careful to protect their staff from radiation hazards. In addition, the military and biomedical applications of the antennas have a major importance due to the proximity of the antenna to the human body. This is why another major concern which is addressed in this study is the influence of the antenna subjected to this study on human tissues. There are a lot of studies regarding this concern. The researchers used the limits imposed by the International Commission on Non-Ionizing Radiation Protection to see how

close the antenna can be to the body for the humans to be safe [2,11,19–27]. Reference [28] was used to determine the radiated power density of the antenna; the distance from the antenna where it is safe for a person to be is also determined. It was discovered that for this case, the antenna should be placed further from human tissues due to the exceeded limits of the standards in force.

The present study is composed of five main sections. The first section is the introduction where some information about the type of analyzed antenna is presented. Section 2 presents the initial structure of antenna which will be optimized, and the tools needed to do that, namely the numerical modeling program which will be used in the study and the Vector Network Analyzer with its software with which the measurements will be conducted. The initial S parameters and gain are determined. Section 2 is also where the antenna is optimized by considering different dielectrics and different geometrical dimensions and positioning of the conductors of the antenna. In Section 3, the Specific Absorption Rate (SAR) and the Radiated Power Density in the vicinity of the structures are analyzed, while in Section 4 the electric and magnetic field values are determined for three layers of different tissues (skin, muscle and adipose tissue) and compared to the limits imposed by the ICNIRP (International Commission on Non-Ionizing Radiation Protection) to determine the safety of using the antenna near the human body. The study is finalized with Section 5, where the conclusions of this study are highlighted.

## 2. Materials and Methods

The initial structure from which this analysis begins is presented in Figure 1, and it is the result of some studies previously published by the authors in [7]. The antenna is constructed on an FR4 epoxy substrate with the relative permittivity of 4.4 and a thickness of 1.51 mm. On one side of the dielectric, there are the 5 directors, while on the other side, there is the reflector. The distance between the feeding dipole and the first director is 5.5 mm, while between the directors, the distance is maintained constant at 3.5 mm. The antenna was constructed with the LPKF ProtoMat S103 Plotter (Figure 2) by importing the geometry designed in the ANSYS High Frequency Software Simulator (HFSS) module to the dedicated design software for the plotter, LPKF Circuit Pro.

After the construction of the structure on the FR4 epoxy substrate, the S parameters were measured with the help of the nanoVNA SAA-2N, and the experimental stand is presented in Figure 3. The parameters can be obtained on the display of the VNA and also using the dedicated software NanoVNA Saver v0.5.3.

In Figure 4, the S parameters obtained after numerically modeling the antenna in Ansys HFSS and the same S parameters measured with the help of the nanoVNA SAA-2N are presented and as expected, they are in accordance. The bandwidth calculated for this structure after considering S11 values obtained through numerical modeling is 15.98%, while that from the measurements is 13.73%.

In Figure 5, the gain of the antenna is presented, and the maximum value obtained from the numerical modeling of this structure is 3.6917. The antenna is linearly polarized.

Beginning from this point, the antenna must be improved from the gain and bandwidth point of view; thus, a number of variations were considered. For a better understanding of this process, a diagram is presented in Figure 6.

It must be stated that the process of reaching an optimum takes a few steps, each step representing a variation of a parameter to reach larger bandwidth and higher gain. Once an optimum structure was reached from this point of view, another parameter is varied. At any time, the better structure is kept as an etalon for the future results and compared to the previous results.

Once an optimum is reached, the authors tried increasing its gain a little bit more while keeping the bandwidth as close as possible to the maximum bandwidth obtained for the optimized structure.

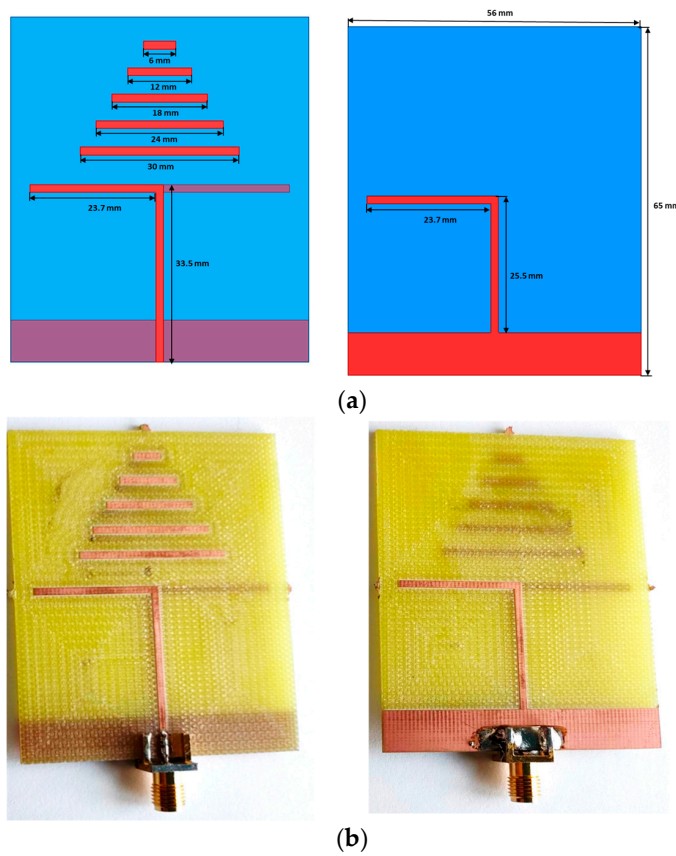

(**a**)

(**b**)

**Figure 1.** The dimensions of the initial antenna structure: (**a**) modeled with HFSS; (**b**) constructed on an FR4 epoxy dielectric.

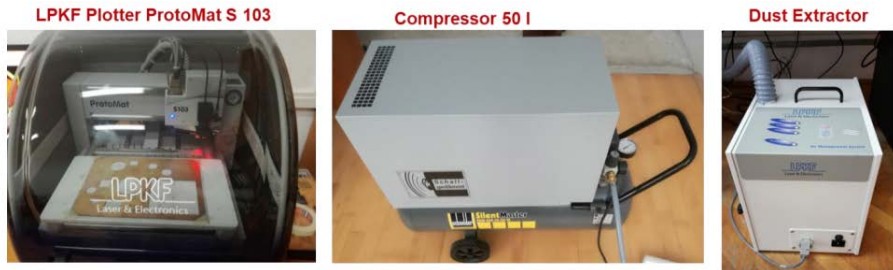

**Figure 2.** The PCB production line LPFK.

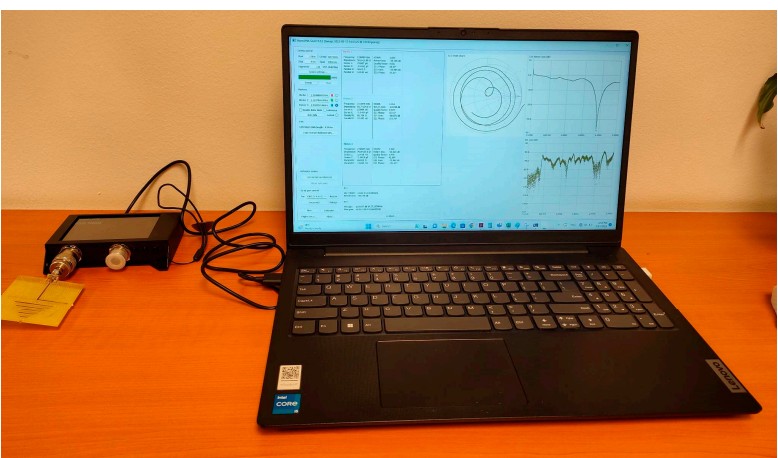

**Figure 3.** Experimental stand for the measurements of the S parameters with nanoVNA SAA-2N.

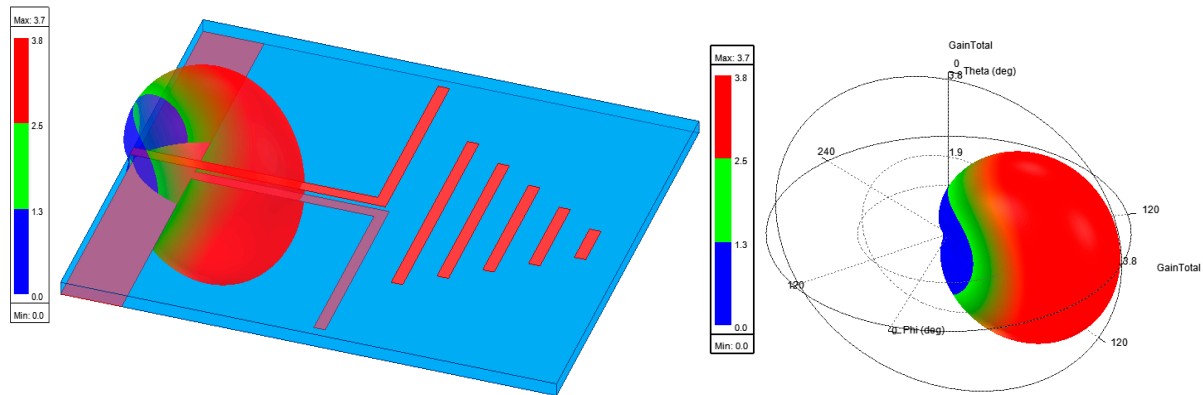

**Figure 4.** S11 values obtained for the initial structure: (**a**) measured; (**b**) obtained through numerical analysis.

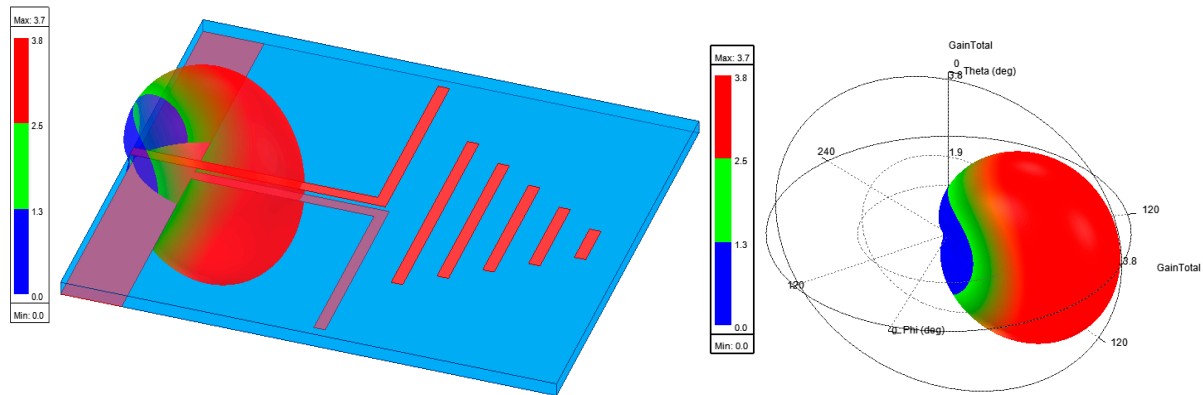

**Figure 5.** Gain of the initial structure.

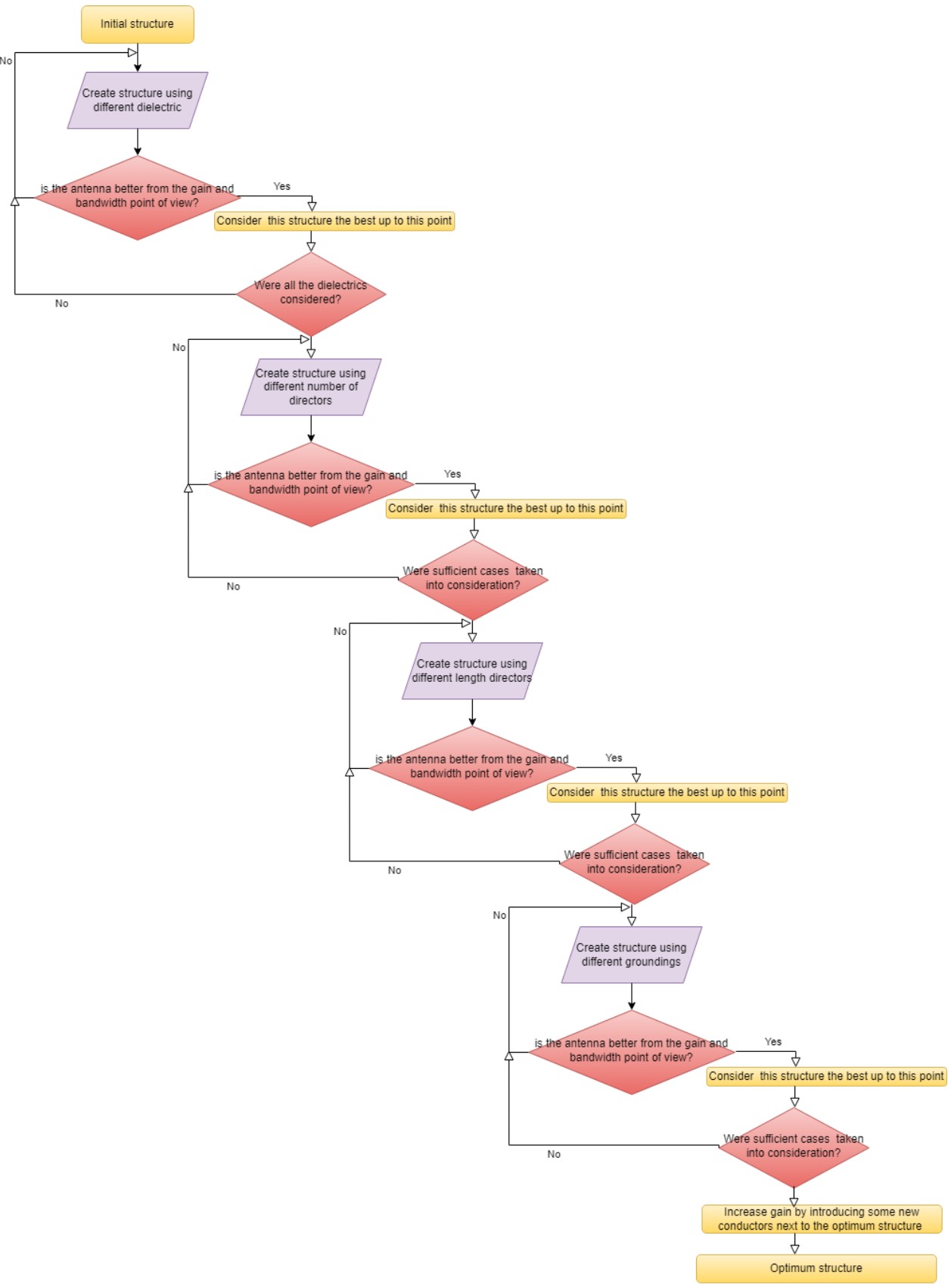

**Figure 6.** Diagram of the optimization process for the planar Yagi Uda antenna analyzed.

### 2.1. Influence of the Dielectric Material on the Antenna Characteristics

Four different dielectric materials were analyzed, FR4 with the relative permittivity of 4.4, RO3003 with $\varepsilon_r = 3$, RO4003C with $\varepsilon_r = 3.55$ and RO4350B ($\varepsilon_r = 3.66$) with two of their standard thicknesses, namely 0.51 mm and 1.51 mm. The S parameters were obtained for the 8 different structures, and the results were represented on 2 different graphs: one for the dielectrics with 0.51 mm thickness (Figure 7a) and the other for the same dielectrics with 1.51 mm thickness (Figure 7b). Even if for the original structure the measurements were made in the frequency range of 1–3 GHz due to the fact that our measurements device is functioning up to 3 GHz, the rest of the numerical modeled structures were analyzed in the frequency range of 1–5 GHz.

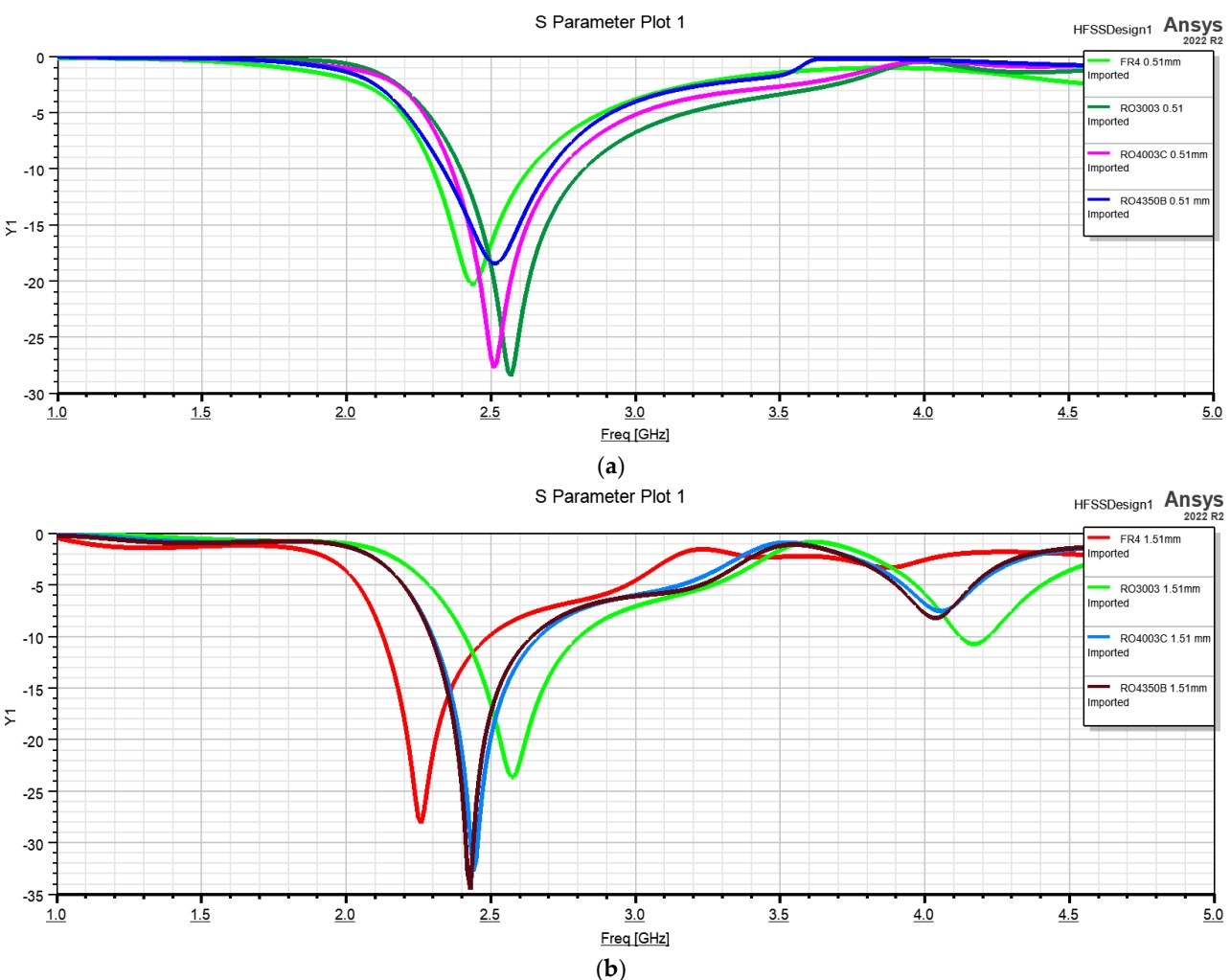

**Figure 7.** S11 values obtained for the structure with different dielectrics with thicknesses: (**a**) 0.51 mm; (**b**) 1.51 mm.

It can be observed that all the analyzed antenna structures have their bandwidth and operating frequency range influenced by their substrate dielectric. The values of S11 are higher for the thinner dielectric in the case of RO3003 and RO4003C, while for a thicker dielectric, S11 has a higher value for RO4350B and RO4003C. Two questions arise after considering these results: the increase in the dielectric thickness influences in the same way the S parameters in all the antennas considered and which structure has the widest bandwidth. In order to answer these questions, the graphs in Figure 8 contain the results for the two dielectric dimensions interpreted for each dielectric type. For each study, it can be stated that the higher S11 values are present for the thicker structures. In addition, the antenna with the dielectric RO3003 of 1.51 mm functions in 2 distinctive bandwidths.

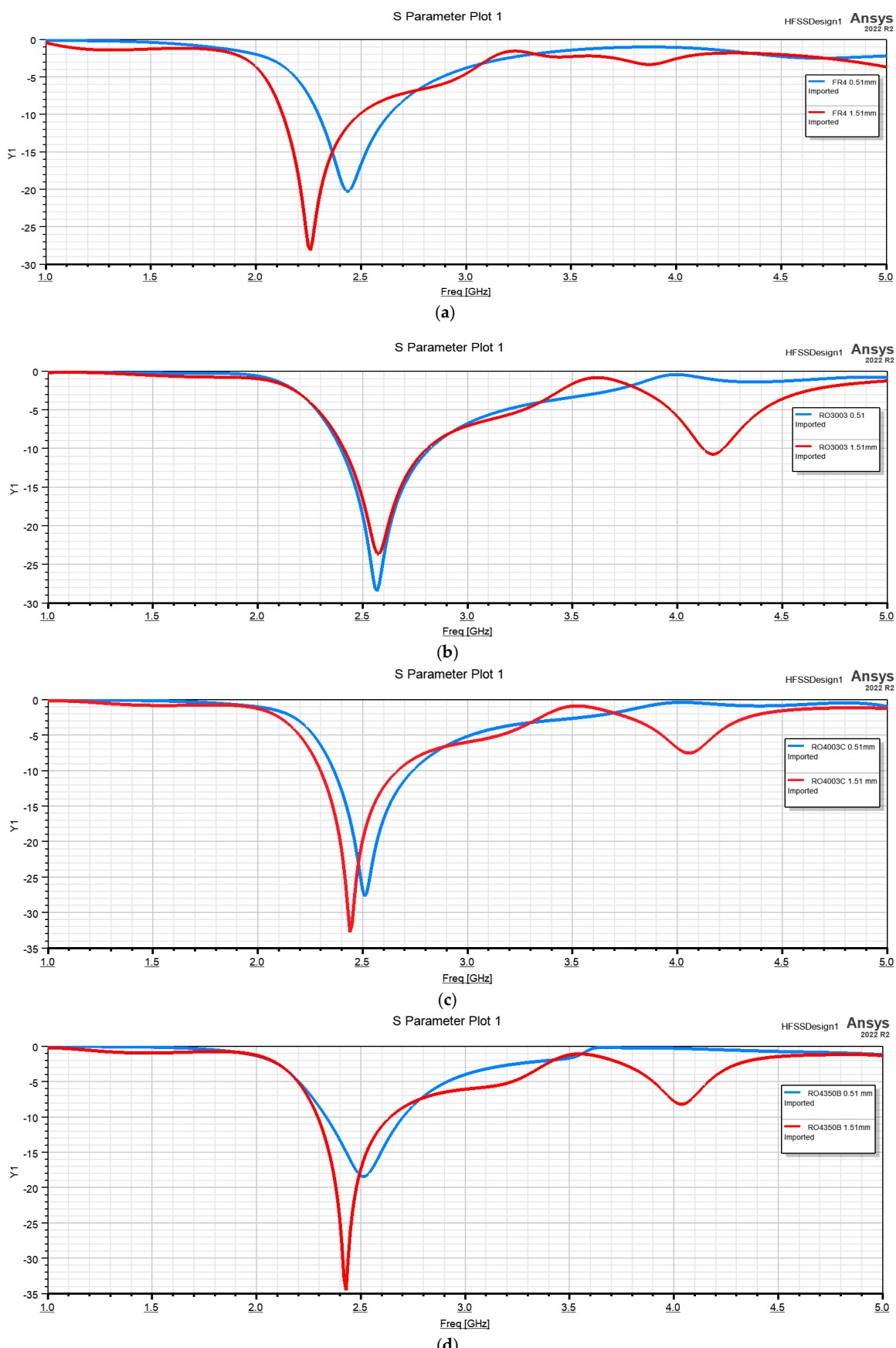

**Figure 8.** S11 values obtained for the structure with different dielectric thicknesses for: (**a**) FR4; (**b**) RO3003; (**c**) RO4003C; (**b**) RO4350B.

The next step is to determine the best configuration from the directivity, front to back ratio and operating frequency point of view, keeping in mind the fact that the frequency for which the antenna should function is 2.4 GHz. The best directivity is obtained for a RO3003 dielectric with 0.51 mm thickness (Table 2). It can be stated that the directivity and gain values decrease for the higher values of the dielectric thicknesses.

**Table 2.** The directivity peak values for the studied structures.

| | Dielectric | | | |
|---|---|---|---|---|
| Dimension | FR4 | RO3003 | RO4003C | RO4350B |
| 0.51 mm | 4.0542 | 4.5328 | 4.46 | 4.37 |
| 1.51 mm | 3.6917 | 4.2514 | 4.2152 | 4.1446 |

The bandwidth on which the analyzed structures are functioning was also determined, and the results can be seen in Table 3.

**Table 3.** The bandwidth for the studied structures.

| | Dielectric | | | |
|---|---|---|---|---|
| Dimension | FR4 | RO3003 | RO4003C | RO4350B |
| 0.51 mm | 13.75% | 16.5% | 14.9% | 14.7% |
| 1.51 mm | 15.98% | 15.32% | 14.9% | 14.2% |

Thus, from the gain and bandwidth point of view, the antenna constructed on a RO3003 dielectric of 0.51 mm thickness is the best, which is followed by the antenna with the same dielectric substrate with a thickness of 1.51 mm. As a compromise between the results, the optimization went further considering this antenna.

### 2.2. Different Number of Directors

For this study, four different cases were considered: namely, structures with 5 directors, 4 directors, 3 close directors and 3 spaced directors (Figure 9). This study was made in order to determine if the dimensions of the antenna could be reduced. Analyzing the S parameters (Figure 10), it can be concluded that the number of directors does not significantly influence the bandwidth of the antenna.

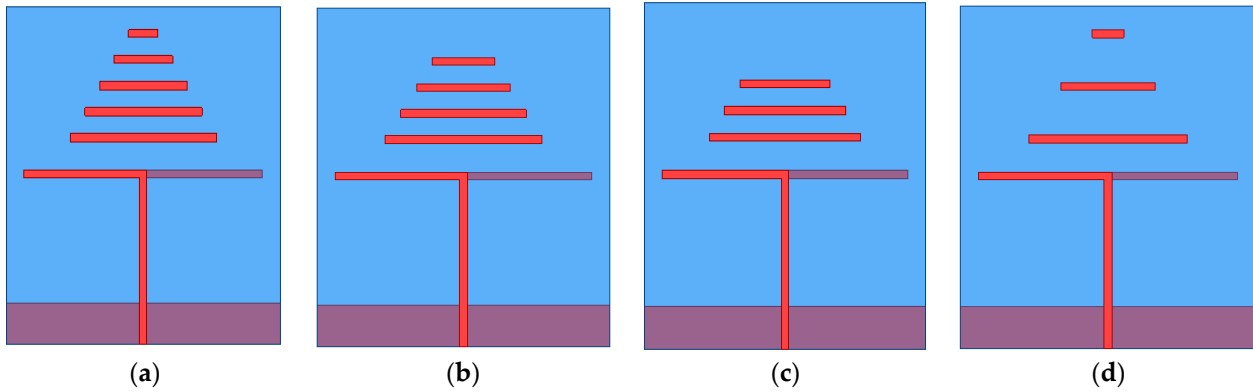

**Figure 9.** The four structures analyzed with: (**a**) 5 directors; (**b**) 4 directors; (**c**) 3 close directors; (**d**) 3 spaced directors.

As it was expected, the gain value is highest for the structures with the higher number of directors. In addition, the front to back ratio is lower when the number of directors is higher; thus, the study will continue with the improvement of the structure with 5 directors.

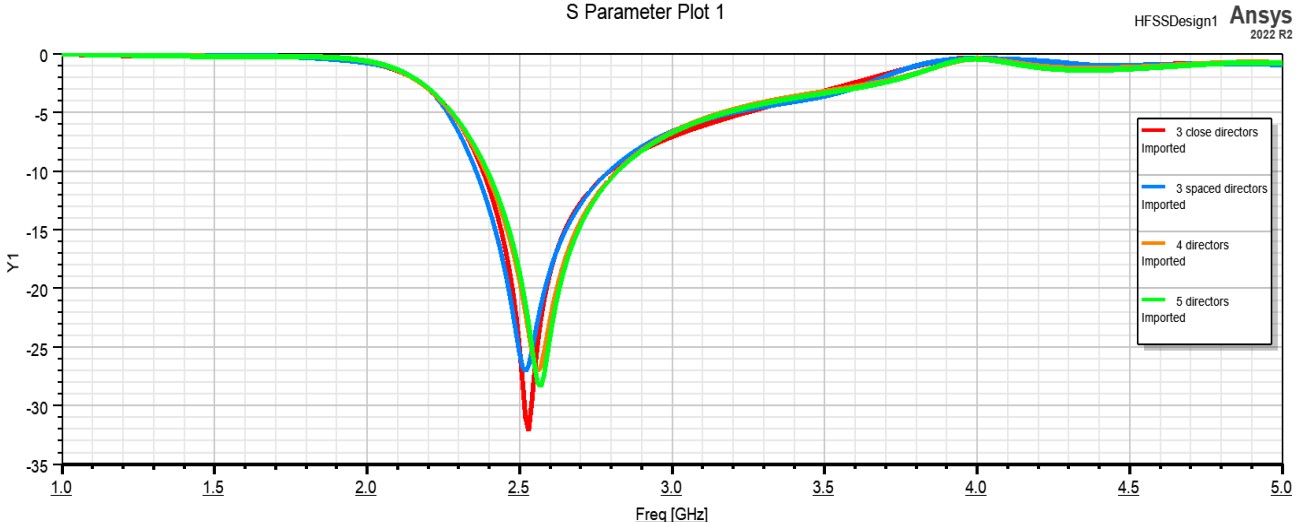

**Figure 10.** Representation of the S parameters for the 4 different structures analyzed.

From the bandwidth point of view, the structure with 3 spaced directors is better, but the difference is negligible; thus, the antenna which will be further optimized is the one with 5 directors.

### 2.3. Different Dimensions of the Directors

The study was continued with the increase in the director's dimensions, in which we increased the director's length by 4 mm, 6 mm, 8 mm and 12 mm. It can be observed that if the antenna's director length is increased by more than 12 mm, the antenna does not function in the considered frequency range.

From the S parameter point of view, the antenna with 5 directors with the increase of 6 mm has the highest bandwidth. The S11 value for this structure has the highest value of them all, −42 dB (Figure 11).

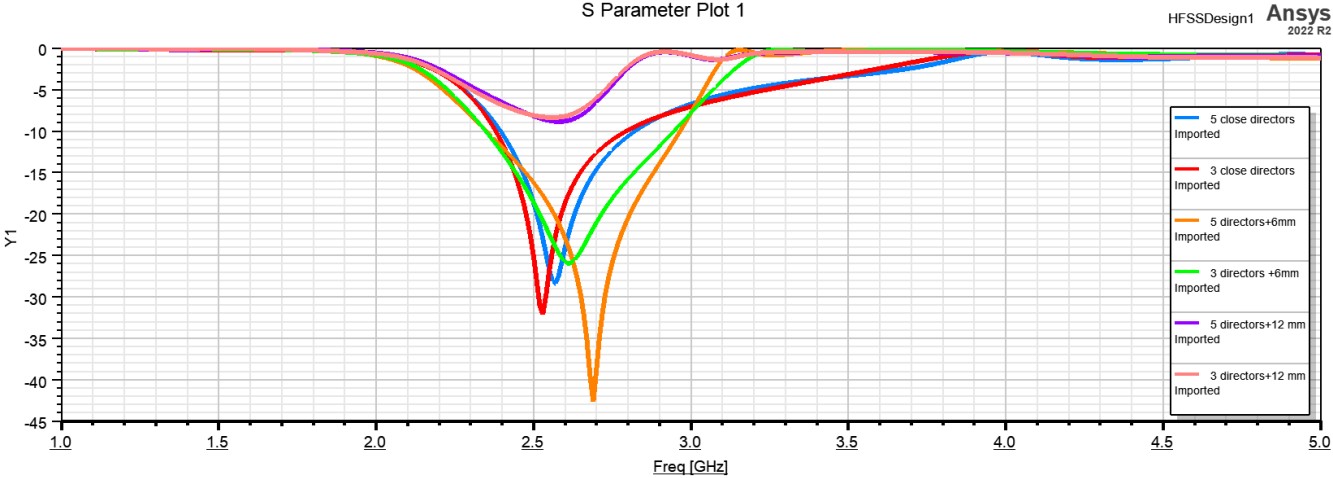

**Figure 11.** Representation of the S parameters for different director lengths.

The gain also increases with the increase in the director's length, the highest value being determined also for the antenna with a 6 mm increase.

Analyzing the bandwidth from the structures without the directors increased with the ones after the increase, there is a major increase determined (Table 4), the bandwidth reaching a value of 23.31% for the structure with 5 directors and an increase of 6 mm in length of the directors: an increase of approximately 7% from the initial structure.

**Table 4.** The bandwidth for the studied structures considering the increase in directors' length.

| 5 Directors | 5 Directors + 6 mm | 3 Directors | 3 Directors + 6 mm |
|:---:|:---:|:---:|:---:|
| 16.51% | 23.31% | 16.22% | 22.64% |

### 2.4. Different Grounding Dimensions for the Antenna

The grounding of the optimum antenna so far was also modified to 4 different lengths to determine its influence. The 5 different structures are presented in Figure 12 along with the grounding for the optimum structure considered so far.

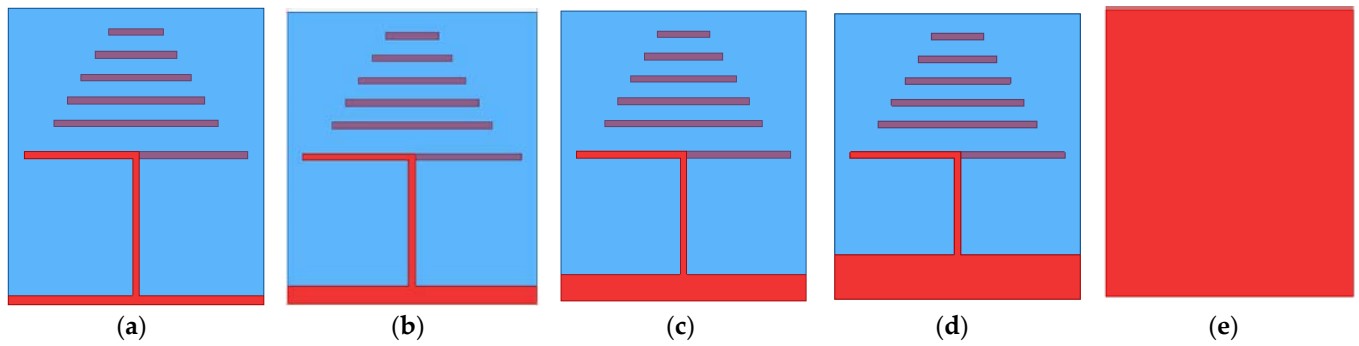

| (**a**) | (**b**) | (**c**) | (**d**) | (**e**) |

**Figure 12.** Structures with different grounding dimensions: (**a**) 2 mm; (**b**) 4 mm; (**c**) 6 mm; (**d**) 10 mm; (**e**) total grounding.

The modification of the grounding influences the bandwidth of the antenna, even increasing it when the grounding has the same dimensions as the dielectric, but the frequency of interest is not in the resulting bandwidth; thus, for this antenna, it is not what we want (Figure 13). In addition, if for the other considered groundings, the gain remains approximately the same, for the full grounding of the dielectric, the value decreases considerably to 0.6 (Table 5). The bandwidths are presented in Table 6, and it can be said that the structure with 8 mm grounding, the one from the previous study, is the best.

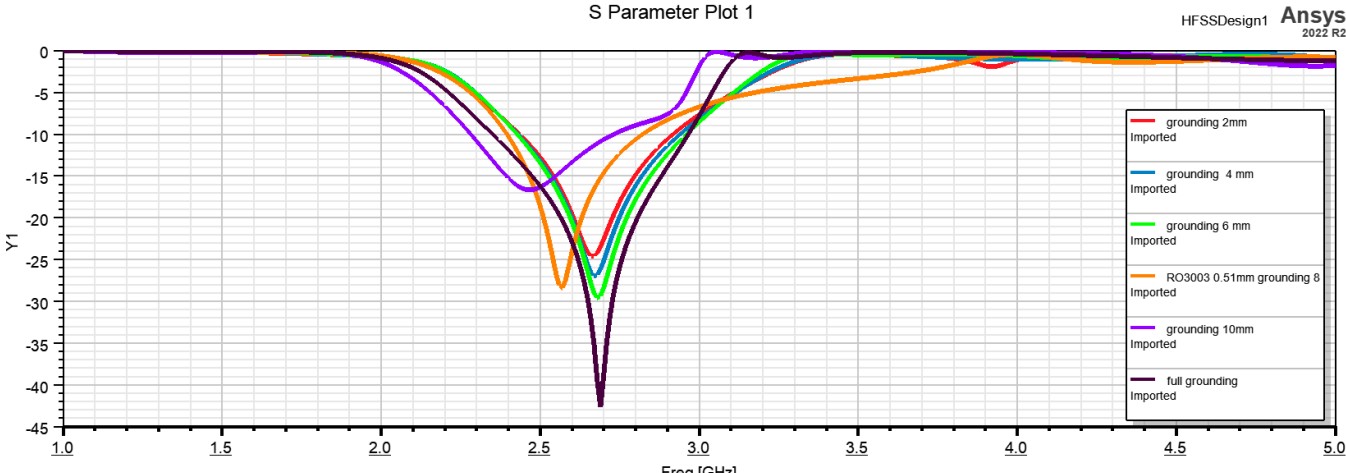

**Figure 13.** S parameters for the structures with different grounding.

**Table 5.** The different peak gain values obtained for the grounding variation.

| 2 mm | 4 mm | 6 mm | 8 mm | 10 mm | Total Grounding |
|:---:|:---:|:---:|:---:|:---:|:---:|
| 4.3219 | 4.587 | 4.673 | 4.5551 | 4.4713 | 0.576 |

**Table 6.** The bandwidths obtained for the grounding variation.

| 2 mm | 4 mm | 6 mm | 8 mm | 10 mm |
|---|---|---|---|---|
| 18.32% | 18.99% | 19.66% | 23.31% | 17.96% |

### 2.5. Improvement of the Optimized Structures Gain

Although the bandwidth was increased along the studies, a final modification is made in the structure aimed at the gain increase. Thus, in Figure 14, the new structure which has 2 microstrip lines along the directors was created. The structure has the characteristics of the optimum determined across all the studies so far; thus, it is made on a Rogers RO3003 substrate of 0.51 mm thickness, with 5 directors with an increased length and a grounding of 8 mm. The measured S parameters in the 1–3 GHz frequency interval are presented in Figure 15a. The results obtained through numerical modeling using the software program HFSS are in accordance with the measured ones obtained using the NanoVNA SAA-2N, but it can be observed that the antenna is functioning immediately after the 3 GHz limit. Thus, in HFSS the frequency sweep was performed between 1 and 5 GHz, and it was seen that indeed, the antenna is functioning on two bandwidths which together give us a 19% bandwidth (Figure 15b).

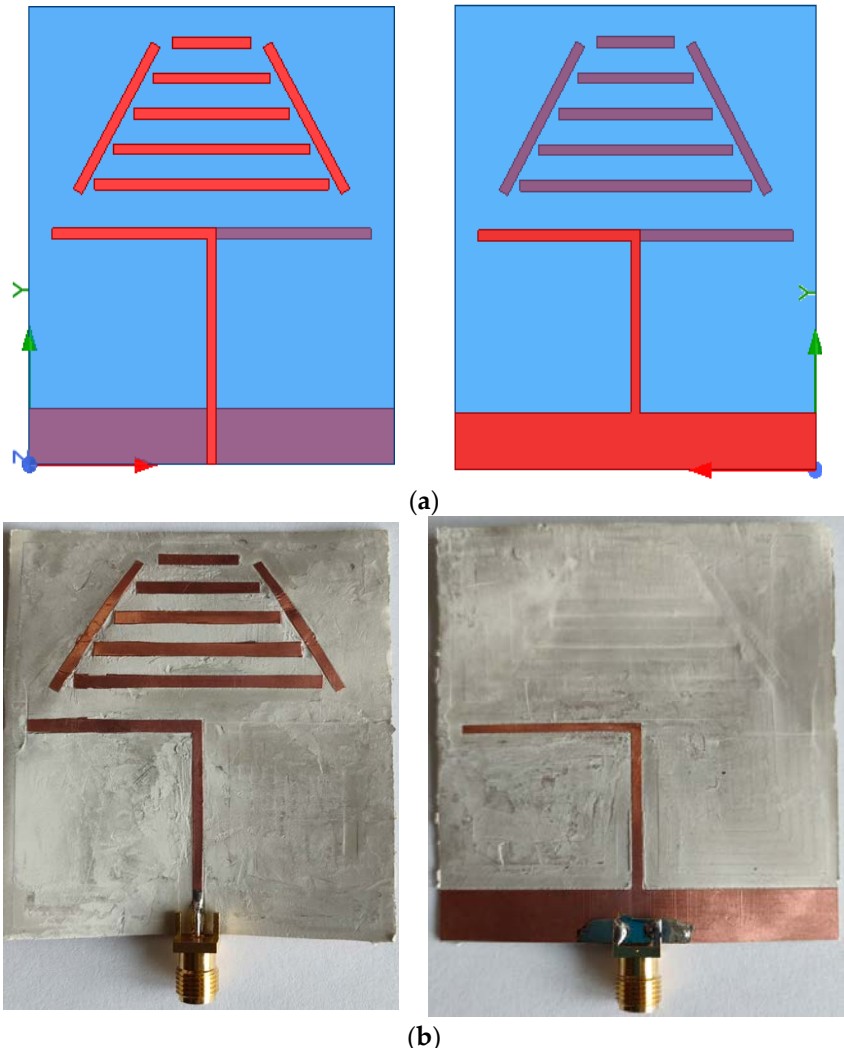

(**a**)

(**b**)

**Figure 14.** The optimized structure: (**a**) modeled with HFSS; and (**b**) constructed on a Rogers RO3003 dielectric.

S11 Return Loss (dB)

**Markers**

| Marker 1 | 2.351851795GHz | 🟥 ⚪ |
| Marker 2 | 2.72226691GHz | 🟩 ⚪ |
| Marker 3 | 2.502961575GHz | 🟦 🔵 |

(a)

S Parameter Plot 1

HFSSDesign1 **Ansys** 2022 R2

dB(S(1,1)) Setup1 : Sweep

(b)

**Figure 15.** S11 values obtained for the optimized structure: (**a**) measured; (**b**) obtained through numerical analysis.

The gain of the structure increased (Figure 16), reaching a value of 4.84, thus increasing the directivity of the original structure by 30%. The dielectric at this thickness is also flexible and thus easy to integrate in close or other systems used for communication at this frequency, but the user must be careful not to fold it because the characteristics of the antenna could be easily changed.

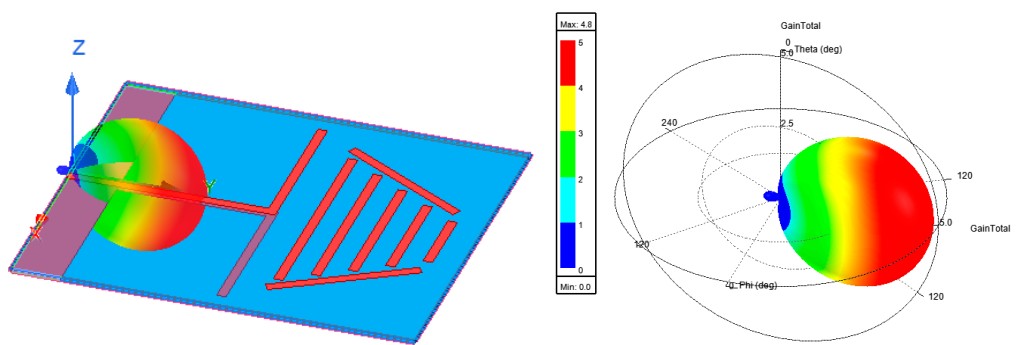

**Figure 16.** Gain of the optimized structure.

### 3. Specific Absorption Rates and Radiated Power Density in the Vicinity of the Structures Analyzed

The values of the specific absorption rate were determined for three types of tissue considered to be close to the analyzed antenna, namely skin, muscle and fat. The values that resulted after numerical modeling of the model are analyzed and compared with the ones suggested by the ICNIRP (International Commission on Non-Ionizing Radiation Protection).

The skin is considered to have a thickness of 4 mm, and the fat is also 4 mm thick, while the muscle tissue is considered to be the widest, 30 mm. The characteristic parameters considered are presented in Table 7. Due to the applications which are mainly situated next to or even on the human tissues, two cases are considered: one where the antennas are placed on the tissue, and the other where the antennas are at a distance of 4 cm from the antennas [19,20].

**Table 7.** Characteristic parameters for the considered tissue types.

| Types of Tissue | | | |
|---|---|---|---|
| | **Relative Permittivity** | **Conductivity** | **Loss Tangent** |
| Skin | 38 | 1.46 | 0.283 |
| Fat | 5.28 | 0.1 | 0.145 |
| Muscle | 52.7 | 1.74 | 0.242 |

The SAR values for general public exposure for more than 6 min is 2 W/kg in the case of the head/torso and 4 W/kg in the vicinity of limbs. For the study, the tissues were considered as seen in Figure 17. A transversal section was performed in the tissue to observe the distribution of SAR in the YZ plane for both the initial and optimized structure, and it can be seen that when the structure is placed on the tissues, the values imposed by the standard are highly exceeded; thus, the antenna must be placed further from the human body.

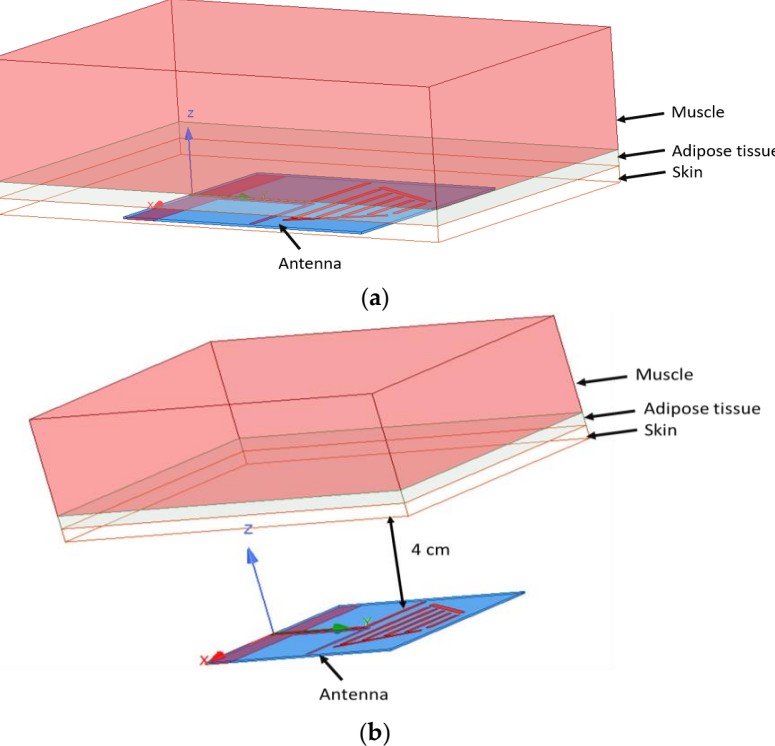

**Figure 17.** Antenna with the 3 different tissues: (**a**) antenna placed on the skin; (**b**) antenna at a distance of 4 cm from the tissues.

In Figure 18, the representation of SAR in the YZ transversal section can be seen. The limits imposed by the standard are exceeded in all the tissue layers for both the initial structure of the antenna and the optimized antenna. The representation of SAR in each of the tissue layers, namely skin, adipose tissue and muscle, can be seen separately in Figure 19. The limits are exceeded in each layer in a wider area for the skin tissue, but they are smaller for the other tissue types.

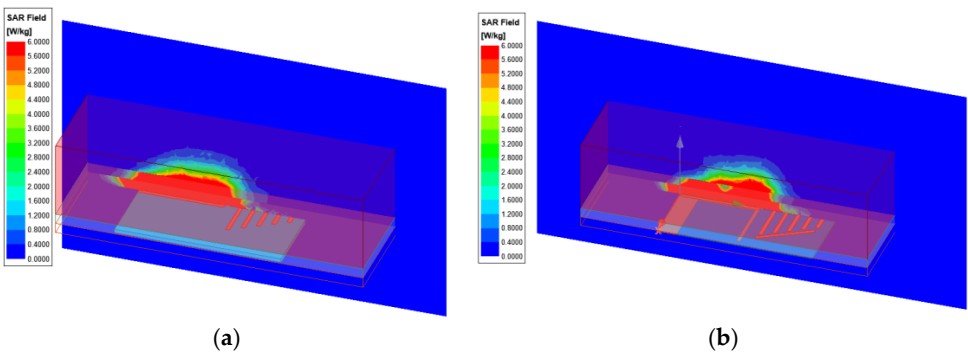

**Figure 18.** SAR distribution on YZ transversal plane for: (**a**) initial antenna; (**b**) optimized antenna.

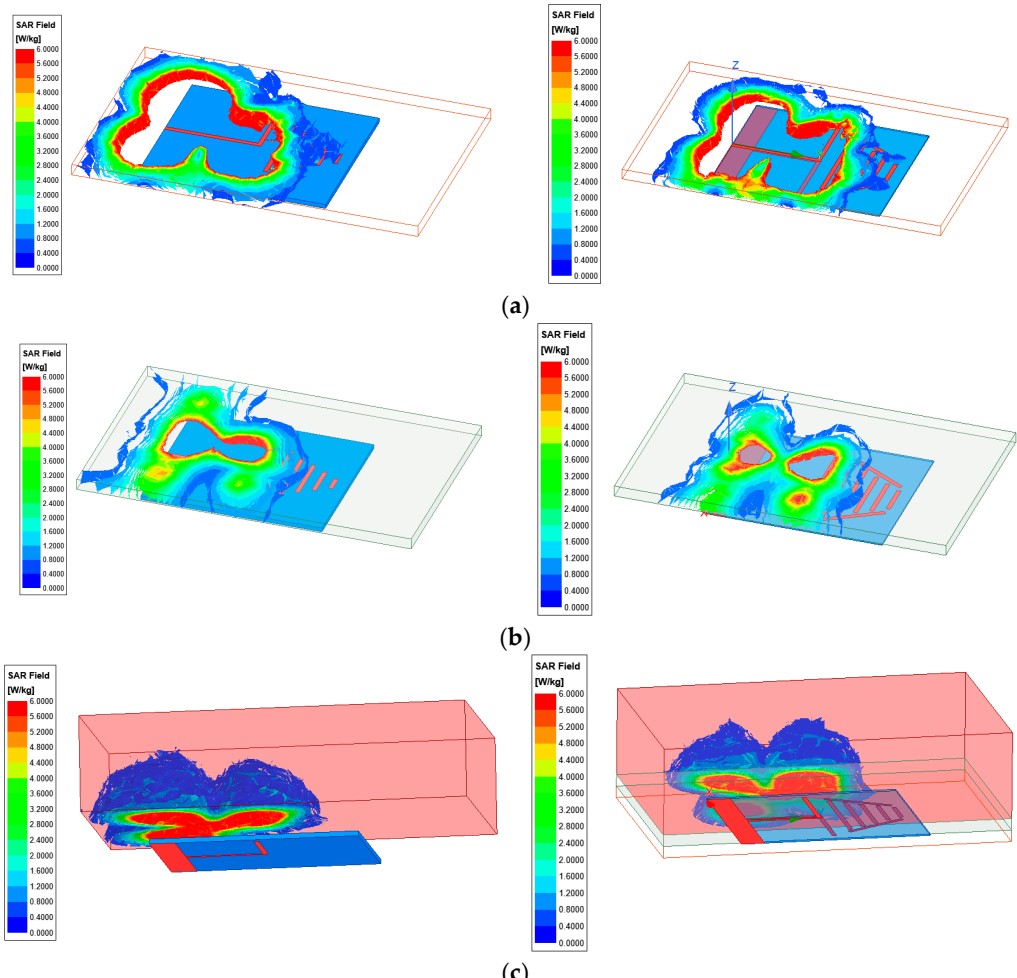

**Figure 19.** SAR distribution in the tissue layers for the initial and optimized antenna: (**a**) skin; (**b**) adipose tissue; (**c**) muscle.

When the tissues are considered to be at a distance of 4 cm, the results are better, the SAR values being exceeded only in the skin tissue; thus, the solution to be in the standard

limits would be to place the antenna at a distance larger than 4 cm from the human tissues, as it can be observed in Figure 20.

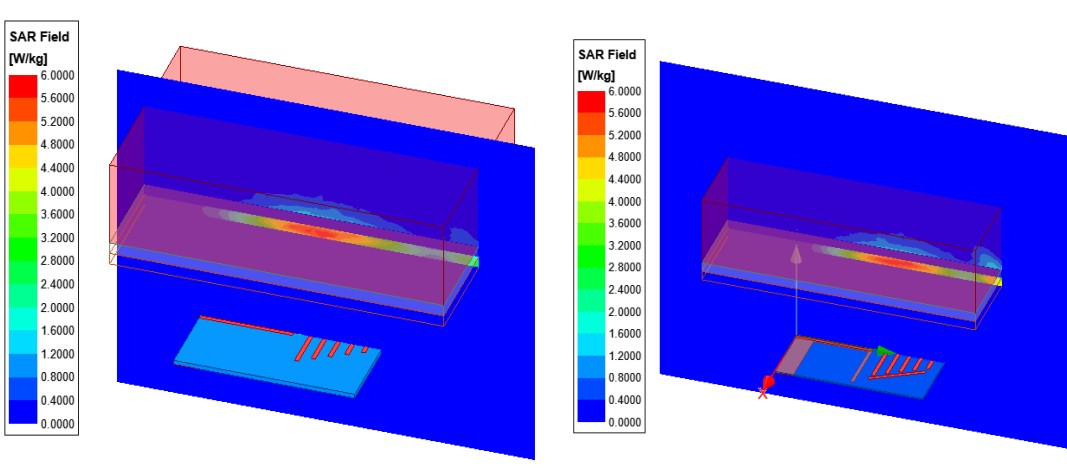

**Figure 20.** SAR distribution on YZ transversal plane for a distance of 4 cm between antenna and tissues: (**a**) initial antenna; (**b**) optimized antenna.

The values of SAR change based on the material characteristics, making it an important parameter that can be used also in applications such as the mapping of snow covering some regions [29], crop mapping [30,31], flood [32] or terrain mapping [33], giving our antenna purposes if the values exceed the ones that are accepted according to the standards in force for the antenna to be in the close vicinity of the human body or embedded in its clothes or near equipment.

According to [28], there are a few reports regarding the limits of the radiated power density of antennas, and there are different limits for different regions or countries. For this study, we had as a guideline the exposure limits from the USA (10 W/m$^2$) and from countries such as Poland, China, Italy and France (1 W/m$^2$), and the radiated power density was calculated with Formula (1)

$$P_d = \frac{P_t \cdot G_t}{4 \cdot \pi \cdot R} \tag{1}$$

where $P_d$ is the radiated power density in W/m$^2$, $P_t$ is the transmitter power in Watts, $G_t$ is the gain of transmitting antenna and $R$ is the distance from the antenna in meters. The results after varying $R$ from 0.01 m to 1 m can be observed in Table 8. It can be determined that the antenna exceeds the limits if the distance between the antenna and the human tissues is less than 0.1 m. If the distance is 0.5 m, the limits are exceeded only for Poland, China, Italy and France, while at a distance of 1 m, the limits imposed are in accordance with the results obtained. Thus, the same conclusions can be drawn: namely, that the antenna must be placed a further from the human body.

**Table 8.** Radiated power density for the initial and optimized antennas.

| R (m) | $P_d$ for the Initial Antenna (W/m$^2$) | $P_d$ for the Optimized Antenna (W/m$^2$) |
|---|---|---|
| 0.01 | 2937.754647 | 3851.540616 |
| 0.05 | 117.5101859 | 154.0616246 |
| 0.1 | 29.37754647 | 38.51540616 |
| 0.5 | 1.175101859 | 1.540616246 |
| 1 | 0.293775465 | 0.385154062 |

## 4. Electric and Magnetic Field Determination in the Vicinity of the Antennas Analyzed

The values of the electric and magnetic fields were determined for both the initial and optimized antenna in order to see if the limits imposed by the standards in force are

in accordance. According to the ICNIRP standard, we can determine the limits for EMF exposure up to 2 GHz, but we considered the same calculation formulas for 2.4 GHz also, considering that the values for the electric field and magnetic field are calculated from the frequency at which the antenna is functioning.

Thus, for an exposure over 30 min, Formulas (2) and (3) are applied, while for an exposure between 6 and 30 min, Formulas (4) and (5) are used.

$$E = 1.375 \, f_M{}^{0.5} \tag{2}$$

$$H = 0.0037 \, f_M{}^{0.5} \tag{3}$$

$$E = 4.72 \, f_M{}^{0.43} \tag{4}$$

$$H = 0.0123 \, f_M{}^{0.43} \tag{5}$$

where $f_M$ is the frequency in MHz for which the antenna is designed.

The maximal values accepted for the frequency of 2.4 GHz are calculated and are presented in Table 9.

**Table 9.** The maximal values accepted for the frequency of 2.4 GHz for the electric and magnetic field.

| Exposure over 30 min | | Exposure between 6 and 30 min | |
| --- | --- | --- | --- |
| E | H | E | H |
| 67.36 | 0.181 | 28.411 | 0.349 |

The electric and magnetic field for the antennas placed directly on the tissue are represented in Figures 21 and 22, considering the legend of the representation between 0 and the maximum values of the electric and magnetic field that are calculated according to the standard ICNIRP for the 2.4 GHz. For both structures, the electric and magnetic field are exceeded in all tissue layers considered.

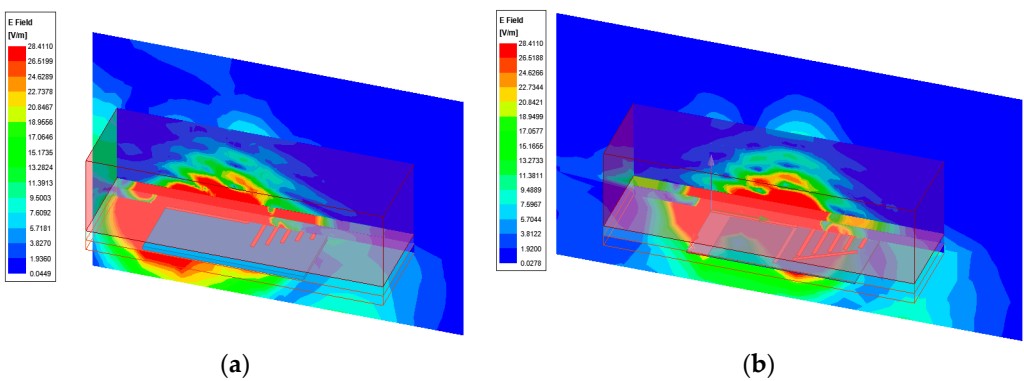

(a)  (b)

**Figure 21.** Electric field distribution on YZ transversal plane: (**a**) initial antenna; (**b**) optimized antenna.

When the tissues are placed at a distance of 4 cm from the antennas (Figures 23 and 24), the values of the electric and magnetic field are exceeding the standard limits for a wider area around the antenna than in the previous case. However, if we consider the tissue layers, it can be stated that the values for the electric and magnetic field have lower values in the tissues if the antenna is placed further from them, as expected; thus, a distance larger than 4 cm from the antenna is needed.

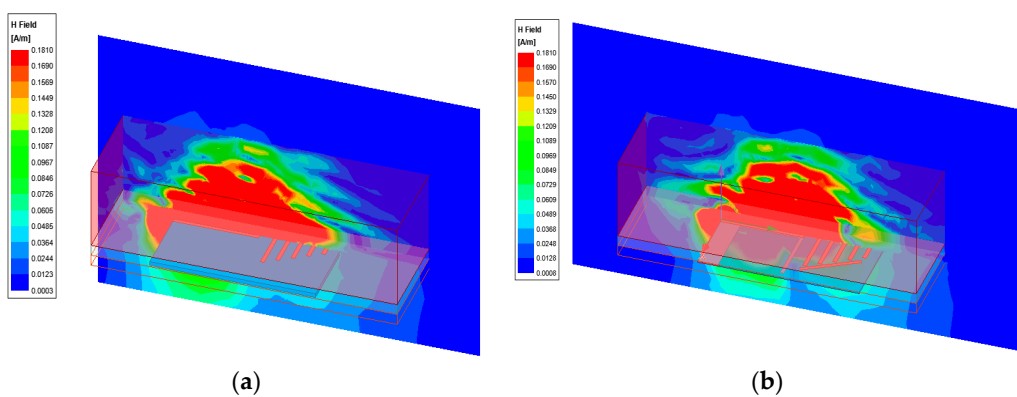

**Figure 22.** Magnetic field distribution on YZ transversal plane: (**a**) initial antenna; (**b**) optimized antenna.

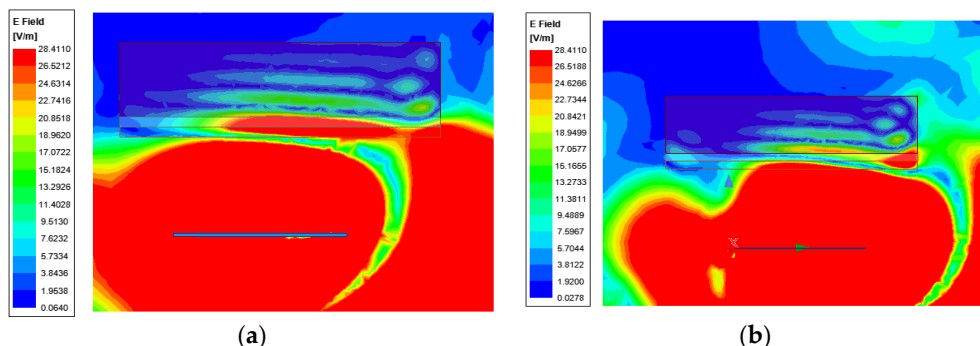

**Figure 23.** Electric field distribution on YZ transversal plane for a distance of 4 cm between antenna and tissues: (**a**) initial antenna; (**b**) optimized antenna.

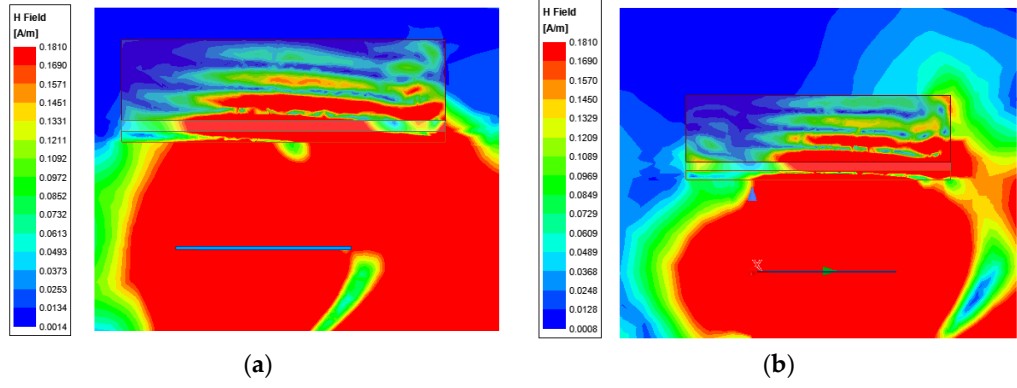

**Figure 24.** Magnetic field distribution on YZ transversal plane for 4 cm between antenna and tissues: (**a**) initial antenna; (**b**) optimized antenna.

## 5. Conclusions

The present study is a complex presentation of the process of optimizing a planar Yagi Uda antenna considering also the effects of placing the resulting structure on the human body. The optimization process consists of the dielectric and geometry dimensions modification while aiming at a larger bandwidth and gain value for the resulting structure. Following the analysis and the data interpretation, the authors reached the conclusion that the optimized antenna has a thinner dielectric with better characteristics, a larger dimension for directors and two more conductor microstrips in order to increase the gain while maintaining the length and width of the initial structure. The results obtained from numerical modeling the proposed structure are compared to the measured results obtained experimentally with a cheaper and smaller VNA present on the market and the results are in accordance, thus determining the fact that the characteristics of antennas can be

preliminary determined in controlled environments with smaller devices with a more easy and cost-effective process.

In the last part of the paper, the authors analyzed the antenna from the human exposure point of view due to the numerous applications of this type of antenna, which is close to the human body. The study was conducted for the electric and magnetic field for radiation power density and also for the SAR values. Following the numerical modeling, it was determined that the antenna analyzed must be placed further from the human body in order for the values of the studied structures to be in accordance with the standards in force.

Throughout the optimization process and the evaluation of the antenna's effects on the human body, the authors determined the steps for the optimization of the antenna, modeled each considered structure in HFSS, and measured the characteristic parameters of the antenna for validation. After reaching an optimum, the authors determined the effects it would have on the human body by modeling and interpreting the results accordingly to the standards in force.

**Author Contributions:** Conceptualization, C.C., C.P., A.G. and C.M.; methodology, C.C., A.G. and C.P.; validation, C.C., A.G., C.P., M.G. and S.A.; formal analysis, C.C.; investigation, C.C., C.P., A.G. and C.M.; resources, S.A. and M.G.; data curation, C.C., C.P., A.G. and C.M.; writing—original draft preparation, C.C.; writing—review and editing, C.C., A.G. and C.P.; visualization, S.A. and M.G.; supervision, C.M.; project administration, C.C.; funding acquisition, C.C. All authors have read and agreed to the published version of the manuscript.

**Funding:** This research received no external funding.

**Institutional Review Board Statement:** Not applicable.

**Informed Consent Statement:** Not applicable.

**Data Availability Statement:** Data sharing not applicable.

**Conflicts of Interest:** The authors declare no conflict of interest.

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
