# Peer review of "High Gain Improved Planar Yagi Uda Antenna for 2.4 GHz Applications and Its Influence on Human Tissues"

_applsci, doi:10.3390/app13116678_

Round 1

Reviewer 1 Report

Comments to the Authors:

The reviewer would like to thank the authors for this thoughtful manuscript. This work has good potential. The authors are requested to put in some additional efforts to improve the quality of this manuscript.

Flowchart in Fig 6. The authors are requested to improve the flow chart. It’s not quite neat and detailed.

Table 1 Can you please present some pictures of the antenna mentioned in this table?

Fig. 3 The authors are requested to present the antenna directivity plot.

Fig. 4 The axis elements in the figure are not visible. Please provide high resolution figures. This is the same with many other figures like Fig 8. Optimization Between Number and Dimension of the Directors.

The authors have not discussed this important point. How do the authors arrive at an optimum point? The authors have indicated optimization of individual factors but a global optimization strategy is missing. Please propose a technique to achieve this

Aspect of Polarization: The discussion around polarization is missing in the manuscript.

Impact of Radiation: The authors are requested to extend the discussion around this topic and cite the following sources.

G. Kumar and N. Kumar, “Report on Cell Tower Radiation”, LAP LAMBERT Academic Publishing, Germany, 2016.

Antennas for Remote Target Detection: The authors are requested to highlight the application of space based antennas of target detection.

Discuss the significance of the proposed research for such applications.

-Muhuri et al., “Snow cover mapping using polarization fraction variation with temporal RADARSAT-2 C-band full-polarimetric SAR data over the Indian Himalayas”, IEEE JSTARS, 2018.

Conclusion: The authors are requested to list the key contributions in this section. At the moment the section is not detailed enough.

The authors have get the article proof read to be convinced.

Author Response

Thank you for your suggestions that helped us improve the quality of our paper. Please see the attachment.

Reviewer 2 Report

The authors have presented a High Gain Improved Planar Yagi Uda Antenna for 2.4 GHz Applications and Its Influence on Human Tissues. This work is exciting. However, there are some comments and concerns that need to be addressed carefully before acceptance. They are as follows:

1- The introduction is insufficient and needs improvement. The authors must provide a more detailed explanation of the techniques used to achieve high gain, considering that the article's title is about high gain antennas [1,2]. Additionally, it would be beneficial if the authors explain the relevance of wearable antennas and provide an explanation of SAR values [3,4]. Please refer to the following articles, which may add value to the introduction:

- A Novel L-Shaped Metalens for Ultra-Wide Band (UWB) Antenna Gain Improvement." Appl. Sci. 2023, 13, 4802. https://doi.org/10.3390/app13084802.

- High gain antenna at 915 MHz for off-grid wireless networks," Bulletin of Electrical Engineering and Informatics (BEEI), vol. 9, no. 6, pp. 2449–2454, 2020, doi:10.11591/eei.v9i6.2192.2020.

- A Miniaturized Full-Ground Dual-Band MIMO Spiral Button Wearable Antenna for 5G and Sub-6 GHz Communications." Sensors 2023, 23, 1997. https://doi.org/10.3390/s23041997.

 -Design of a Tri-Band Wearable Antenna for Millimeter-Wave 5G Applications." Sensors 2022, 22, 8012. https://doi.org/10.3390/s22208012.

2- Please improve Table 1 and add more recent papers to confirm the novelty of the paper.

3- The sentence in lines 74 and 75 is awkward. Please rephrase it for better clarity.

4- There are duplicates in Figure 10, with two instances of Figure 10 in the paper. Please rectify this duplication.

5- Figures 6, 7, 8, 10, 11, and 12 are not clear. The authors have drawn these figures using a traditional method in Microsoft Excel. It is recommended to redraw the figures using a more scientifically acceptable method.

6- The SAR values for general public exposure must be added in the abstract section.

7- The paper should include the simulated and measured gain.

8- The paper should include the simulated and measured efficiency.

9- The paper should include the simulated and measured radiation pattern.

Addressing these comments and concerns will greatly enhance the quality and impact of the research. 

That's all for me at this moment! The authors are required to revise the comments above carefully. Thanks

There are minor errors that need to be carefully checked!

Author Response

(The authors gave the same response as above.)

Reviewer 3 Report

1- The abstract needs to be reformulated to include all results

2- The figures 7,8,10,12 .. the graphics need to be redrawn in order to align with the rest as the curves are displayed

3-Figure 10 numbering is repeated

4- the paper titled with high gain, however Gain -frequency curves is not mentioned

5- front back ratio in dB with frequency is needed 

6- Some curves are missing axis identification please correct it 

Author Response

(The authors gave the same response as above.)

Round 2

Reviewer 2 Report

The authors made significant efforts in revising the manuscript, and the paper has improved. However, there is still no measurement provided for gain, efficiency, and radiation pattern. The authors claim that they lack the necessary equipment in their lab. If this is acceptable by the academic editors, I am also fine with it.

Best regards,

Moderate editing of English language required!